# A sensitive tissue factor activity assay determined by an optimized thrombin generation method

**Søren Risom Kristensen**[1,2]*, **Jette Nybo**[1]

**1** Department of Clinical Biochemistry, Aalborg University Hospital, Aalborg, Denmark, **2** Department of Clinical Medicine, Aalborg University, Aalborg, Denmark

* srk@rn.dk

## Abstract

### Background

Tissue factor (TF) is the principal activator of the coagulation system, but an increased concentration in the blood in cancer and inflammatory diseases has been suggested to play a role increasing the risk of venous thromboembolism. However, measurement of the TF concentration is difficult, and quantitation of activity is the most valid estimation. The objective of this study was to establish a sensitive method to measure TF activity based on thrombin generation.

### Methods

The assay is based on thrombin generation (TG) measured on the Calibrated Automated Thrombogram (CAT). Various low concentrations of TF were prepared from reagents containing 1 pM TF and 4 μM phospholipid (PPL), and no TF and 4 μM PPL, and a calibration curve was produced from Lagtime vs TF concentration. TF in blood samples was measured after isolation and resuspension of extracellular vesicles (EVs) in a standard plasma from which EVs had been removed. The same standard plasma was used for the calibrators.

### Results

Contact activation of the coagulation system was avoided using CTI plasma samples in Monovette tubes. EVs contain procoagulant phospholipids but addition of PPL only reduced lagtime slightly at very low concentrations of TF resulting in overestimation to a lesser extent at 10 fM but no interference at 30 fM or higher. Addition of EVs to the TG analysis induced a small unspecific TF-independent activity (i.e., an activity not inhibited by antibodies against TF) which also may result in a smaller error in estimation of TF activity at very low levels but the effect was negligible at higher concentrations. It was possible to measure TF activity in healthy controls which was found to be 1–6 fM (EVs were concentrated, i.e. solubilized in a lower volume than the original volume plasma). Coefficient of variation (CV) was below 20% at the low level, and below 10% at a level around 100 fM TF. However, the step with isolation of EVs have a higher inherent CV.

**Data Availability Statement:** All relevant data are within the paper.

**Funding:** The authors received no specific funding for this work.

**Competing interests:** The authors have declared that no competing interests exist.

## Conclusion

A sensitive and rather precise one-stage TG-based method to measure TF activity has been established.

## Introduction

Tissue factor (TF) is the primary coagulation factor that initiates activation of the coagulation system which leads to fibrin formation [1]. This is crucial to avoid bleeding after traumas, but an increased formation has also been ascribed a role in the increased risk of venous thromboses (VTE) in e.g. cancer and inflammatory diseases [2, 3]. TF is present in all vascular walls and is normally absent or present in very low concentrations in plasma [4, 5]. However, cancer cells and monocytes may synthesize TF and release extracellular vesicles (EVs) containing TF as a transmembranal protein (EV-TF), and this may contribute to thrombosis [2, 6]. Therefore, measurement of TF is interesting in order to describe a potentially increased risk of VTE in patients.

However, determination of tissue factor is difficult. It can be measured as the antigenic level, but these methods measure very different levels including non-active forms [4, 7]. Therefore, activity assays are preferred as more reliable tests. The probably most used method is a two-stage method where EVs are isolated with a centrifugation at 20,000 g for 15 min [8, 9] (later improved with a centrifugation step for 60 min [10]) and incubated with Factor X (FX) and FVIIa [8, 9] or FVII [10], and the TF activity in combination with FVIIa will activate FX to FXa; in the second stage a chromogenic substrate is added to quantitate FXa [8–10]. The assay is performed in the presence and absence of antibodies against TF to ensure specificity of TF-driven formation of FXa. Several other methods as one-stage quantitations or use of antibody-capture of EVs have been described, but according to Mackman et al. [7] none of these methods are neither as sensitive nor specific to quantitate TF as the two-stage method mentioned above. Østerud et al. [5] indicated very recently, however, that this method is not completely specific, mainly because of a long incubation time where FXa itself may activate FX, thus not specific for TF activity. In the paper of Østerud et al. a new sensitive and specific method was described [5]. It has a similar principle but includes also addition of FV and procoagulant phospholipids (PPL) and prothrombin, and the generation of thrombin is followed using a chromogenic substrate. This reduces the incubation time to a few minutes.

The thrombin generation assay (TG), Calibrated Automated Thrombogram (CAT) developed by Hemker and colleagues [11, 12] has also been demonstrated to be able to detect TF in several papers [13–16]. Ollivier et al. [13] showed that especially lagtime depends on the TF concentration, i.e. a higher concentration of TF reduces the lagtime. However, an optimized method measuring low levels of TF has not been described. The aim of the present study was to establish a sensitive one-stage method to determine TF activity based on the CAT method. The principle is to isolate EVs from plasma which are added to a standard plasma where contact activation has been eliminated making the assay fully dependent on activation from TF activity. To optimize the method, we examined various important factors and additions to make it sensitive and specific for TF activity. Linearity was demonstrated and variation of the optimized method was estimated. Furthermore, we tested the effect of addition of procoagulant phospholipids, which also is present in EVs.

## Materials and methods

### Materials

Reagents for TG measured by the CAT method (Thrombinoscope BV, Maastricht, The Netherlands) were MP-reagent containing 4μM phospholipids (PL), PPP-reagent Low containing 1

pM TF and 4 μM PL, and Fluca-buffer containing a fluorescent substrate (Z-Gly-Gly-Arg-AMC) and CaCl₂ (Stago, Thrombinoscope BV, Netherlands). Corn Trypsin Inhibitor (CTI) and antibodies against TFPI (aTFPI) (Sheep Anti-Human Tissue Factor Pathway Inhibitor) were from Haematologic Technologies. Antibodies against TF (aTF) (HTF-1, Purified Mouse Anti-Human CD142) were from BD Biosciences. Antibodies against FXII (aFXII) were from Sanquin, Amsterdam, the Netherlands (clone CLB OT-2). Lipopolysaccharides (LPS) and soybean trypsin inhibitor were from Sigma-Aldrich, St. Louis.

## Experiments

The method was developed successively by first testing surface activation, i.e. effects of sample tube (Vacutainer and Monovette), and possible inhibition of surface activation by adding antibodies against FXII, and of addition of CTI. TFPI binds to and inactivates TF or rather TF: FVIIa:FXa complex reducing the signal from TF [17], and we therefore tested addition of antibodies against TFPI (aTFPI) in these experiments in order to maintain the TF activity in the sample, i.e. increase the signal. A concentration of 100 μg/mL has been used previously [16], and the effect of a higher concentration was tested to ensure complete inactivation. Addition of antibodies against TF was used to ensure that the measured activity is TF-activity; a concentration published previously [18] was used, and higher concentrations were tested to ensure full TF inhibition. Since TF is cofactor to FVIIa which activates FX to FXa, we tested whether addition of FVIIa could increase the signal. EVs contain procoagulant phospholipids (PPL). The effect of increasing the concentration of phospholipids was tested to make sure that this will not affect the measured TF activity. With the established method, linearity was demonstrated and variation calculated, measuring several samples with various concentrations including EVs from 7 normal individuals and also plasma from blood activated with LPS which increases the TF activity.

## Blood samples

To prepare standard plasmas (SP) for the assay: Blood was sampled from healthy volunteers which included 5–10 persons; it was performed several times during the period and the number was adjusted after the planned series of experiments. After oral and written information and signed consent (the study was approved by the local Ethical Committee, N-20220011) blood was sampled using a 21-gauge needle in 3.2% (w/v) trisodium citrate. The first tube was discarded. The samples were centrifuged twice at 2500*g and 20˚C in 15 min as recommended [6]; subsequently plasma was centrifuged at 20.000*g in 1 hour at 4˚C to remove EVs, and the supernatants were pooled. In the initial experiments blood from Vacutainer tubes (BD, Plymouth, UK) were used but since these tubes have a rather high level of surface activation, we changed to Monovette tubes (Sarstedt, Nümbrecht, Germany). After testing for the effect of CTI, CTI was added to the Monovette tubes (18,3 μg CTI per mL citrated whole blood as suggested by Luddington and Baglin [19]).

Samples for measurement of TF: Blood samples were centrifuged twice at 2500*g and 20˚C in 15 min [20], and EVs were isolated after centrifugation of plasma at 20.000*g for 1 hour at 4˚C; the sediment was washed with phosphate buffered saline and centrifuged once more at 20.000*g in 1 hour at 4˚C. The final sediment was resuspended in SP with aTFPI added (100 μg/mL).

LPS stimulation of blood samples: 10 μg/mL LPS were added to the blood sample which was incubated at 37˚C for five hours. Following that, the samples were centrifuged and EVs isolated as described above.

In seven healthy volunteers TF activity was measured in EVs from plasma and from plasma from LPS-stimulated blood. The volunteers included 6 women and 1 man, age 25–68 years.

## Thrombin generation

For all experiments were used SP to which aTFPI was added (100 μg/mL plasma) and incubated 15 min at 37°C before use. TG measurements were conducted according to the manufacturer's instruction using 80 μl sample mixed with 20 μl of trigger solution (i.e., calibration curve: 80 μL SP + 20 μL dilution of TF (dilution of PPP-reagent Low with MP-reagent to get the various concentrations of TF); samples: 60 μL SP + 20 μL EVs suspension (in SP) + 20 μL MP-reagent), and subsequently coagulation was initiated by the addition of 20 μl FluCa-buffer. All measurements were performed in duplicates. Each plasma measurement was calibrated against the same plasma mixed with 20 μl Thrombin Calibrator (Thrombinoscope BV, Maastricht, The Netherlands). Fluorescence of AMC (7-amino-4methylcoumarin) was measured on a Fluoroskan Ascent (Thermo Scientific, Waltham, MA, USA), 390 nm excitation and 460 nm emission. The dedicated Thrombinoscope software (Thrombinoscope BV) was used to calculate: Lagtime (LT), Endogenous thrombin potential (ETP), Peak, and time to Peak of which mainly LT was used in this study. All samples were run without and with addition of aTF (7,84 μg/mL) incubated 15 min at 37°C before analysis to ensure inhibition of TF.

Trigger solutions: The calibration-curve was constructed from mixtures of the trigger solutions PPP-reagent Low and MP-reagent in various ratios to make solutions with TF concentrations from 0–0.1 or 0.5 pM TF (dependent on the range of expected TF concentrations). For measurement of TF concentrations, the solutions of isolated EVs in SP were used. When examining the effect of an increased concentration of PPL, the MP-kit reconstituted with half volume water was used (0 pM TF, 8 μM PPL) to achieve PPL concentrations of 5–8 μM.

The coefficient of variations (CVs) of measurements are described at the end of the Results section.

## Statistical analysis

All experiments were performed several times using duplicates, and representative experiments are shown in the figures. Graph Pad was used to construct the figures. To test for differences in the various tests of different concentrations etc., a Students t-test was used.

Imprecision was determined as coefficient of variation (CV) i.e., the standard deviation of the measurements of each sample divided by the mean, and the pooled CV was calculated as the square root of the mean of the squared CVs.

## Results

When measuring low levels of TF it is important to avoid contact activation, and therefore the choice of sample tube is important. Fig 1A shows thrombin generation in plasma sampled in Monovette tubes and Vacutainer tubes, respectively. Contact activation is obviously higher in Vacutainer tubes, but it is also present in Monovette tubes at a lower level. We, therefore tested addition of antibodies against FXIIa [18, 21] and addition of corn trypsin inhibitor (CTI) to inhibit the contact activation (S1 Fig). LT increased in the presence of aFXII, but after addition of 18.3 μg/L CTI [19] LT increased much more indicating a high degree of inhibition of the contact activation. Using Monovette tubes with addition of CTI (Fig 1B) virtually no thrombin was generated if no TF was added, and in samples with a low concentration of TF the thrombin generation was totally inhibited by antibodies against TF (aTF). A complete inhibition of aTF at this concentration was ensured by testing a double and triple concentration (S2 Fig). A higher concentration of CTI may be needed for complete inhibition of FXII [22, 23], and

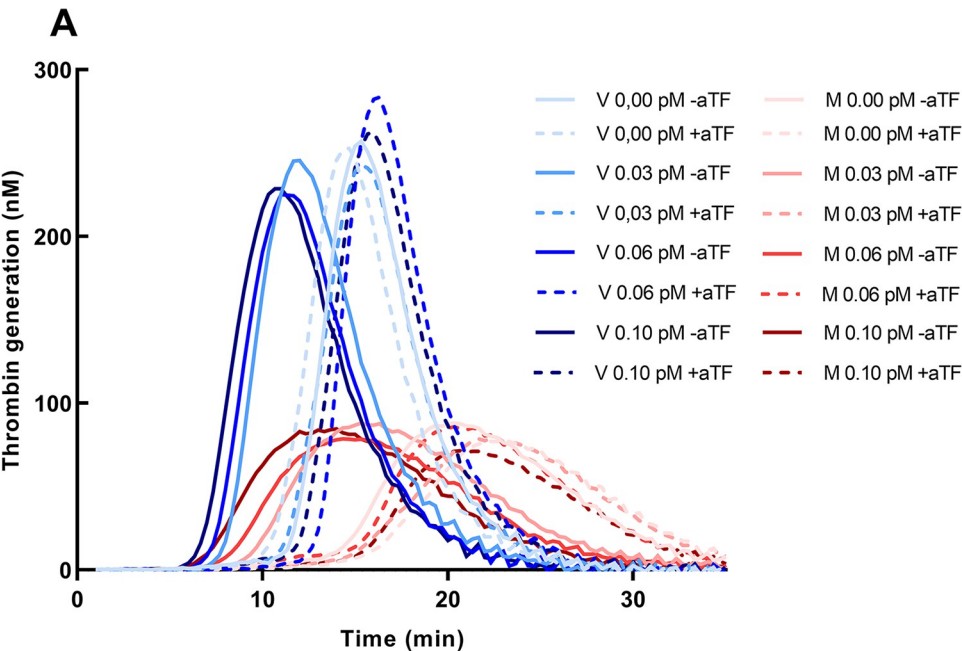

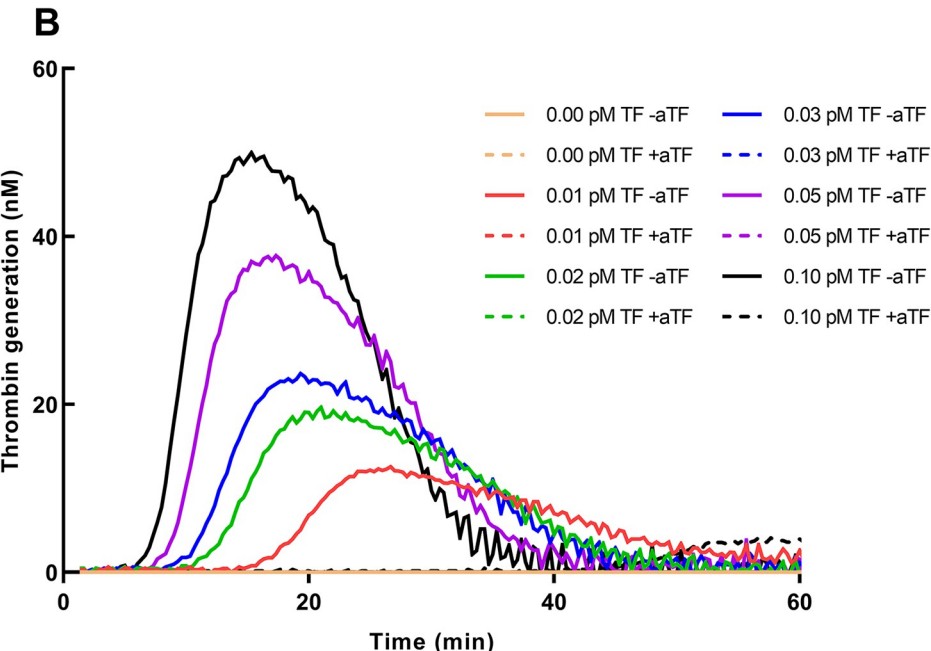

**Fig 1. Thrombin generation in Vacutainer and Monovette tubes, and Monovette tubes after addition of CTI.** A shows the thrombin generation at low TF concentrations in plasma from Vacutainer tubes (blue shades) and Monovette tubes (red shades). The dashed lines are samples with antibodies against TF (aTF). B shows the thrombin generation at low TF concentrations in samples from Monovette tubes with 18.3 µg/mL corn trypsin inhibitor (CTI) added.

consequently we tested and compared addition of 18.3 and 36.6 μg/mL CTI (S3 Fig). A slightly longer LT was found in samples with 36.6 μg/mL, but the differences were minimal resulting in almost the same levels of TF concentration, and addition of 18.3 μg/L was chosen. TFPI will inactivate some TF and therefore antibodies against TFPI (aTFPI) was added in a concentration of 100 μg/mL [16] which increased the signals of the low TF concentrations (shorter LT) with a large difference between LT in the presence and absence of aTFPI (S4 Fig)). A higher concentration of 150 μg/L did not increase the inhibition, and consequently, we used a concentration of 100 μg/mL in the following experiments. The signals also had a lower variation of the duplicates after addition of aTFPI.

It was tested whether addition of FVIIa could improve the signals since the activity of TF is performed by the complex between TF and FVIIa. High concentrations of FVIIa to as much as 1000 pM had an effect on LT which decreased showing an apparent higher TF activity. However, an effect was also seen in samples with aTF added, indicating that this effect was not through complexing with TF (S5 Fig). Consequently, this addition was not beneficial and was not used in the following experiments.

Using the final design with Monovette blood with CTI added and addition of aTFPI to plasma, a calibration curve with TF concentration vs LT was linear in a log-log system, Fig 2. A calibration curve to determine the TF concentrations was created in all the following experiments, always with the same appearance.

Linearity of the method was ensured measuring dilutions of plasma from LPS stimulated blood (an example is shown in Fig 3).

To measure TF in a plasma sample, EVs were isolated by centrifugation at 20,000*g for one hour and resuspended in a lower volume SP to increase the concentration of TF (dependent on the expected concentration). TF concentrations of EVs in healthy persons were measurable if they were concentrated, i.e. EVs from 1500 μl plasma (or higher) were resuspended in 300 μl SP (Fig 4A) before mixing with SP in the TG analysis. The levels in 7 healthy controls were between 1 and 6 fM (TG curves for all 7 individuals are shown in S6 Fig). Higher concentrations were achieved in blood samples incubated with LPS (Fig 4B). The undiluted levels of TF in EVs from plasma from LPS stimulated blood were between 100 and 800 fM, where EVs from 500 μl plasma were resuspended in 300 μl SP. The samples with EVs showed a small TG activity in samples with aTF added (Fig 4A and 4B), i.e. an unspecific TF-independent activity, possibly surface activation by the EVs in spite of addition of CTI. This was most marked in samples with a low TF activity with a concentrated amount of EVs in the TG sample. It has been suggested that a similar effect of EVs from red blood cells can be inhibited by soybean trypsin inhibitor (SBTI) [24]. We, therefore, tested the effect of SBTI (S7 Fig). At a concentration of 1 μg/mL there was no effect. Higher concentrations of SBTI inhibited the TF activity as well as activity in the presence of aTF, and, therefore, we could not use this addition. In the calculation of TF activity from the calibration curves (Fig 2) we calculated the concentration of TF activity in the samples without aTF as well as with aTF and subtracted the apparent concentration in samples with aTF added. The premise for this calculation is that the activations by TF activity and the unspecific TF-independent activity are additive which may not be true. But this unspecific TF-independent activity was always substantially smaller than the concentration without aTF. For the low values it was 10–25%, mainly 15–20%, of the measured activity, but for the higher activities above 100 fM, it was less than 1%. Thus, this TF-independet activity did not interfere much with the result although not negligible at the low level.

EVs contain procoagulant phospholipids (PPL) as well as TF, and, therefore, the content of PPL may affect TG. The effect of higher concentrations of PPL was tested by addition of 1, 2 and 4 μmol/L PPL to the trigger solution (i.e, final concentrations of 5, 6 and 8 μmol/L). Fig 5 shows that although TG increased at a higher PPL concentration with an increased ETP and

**A**

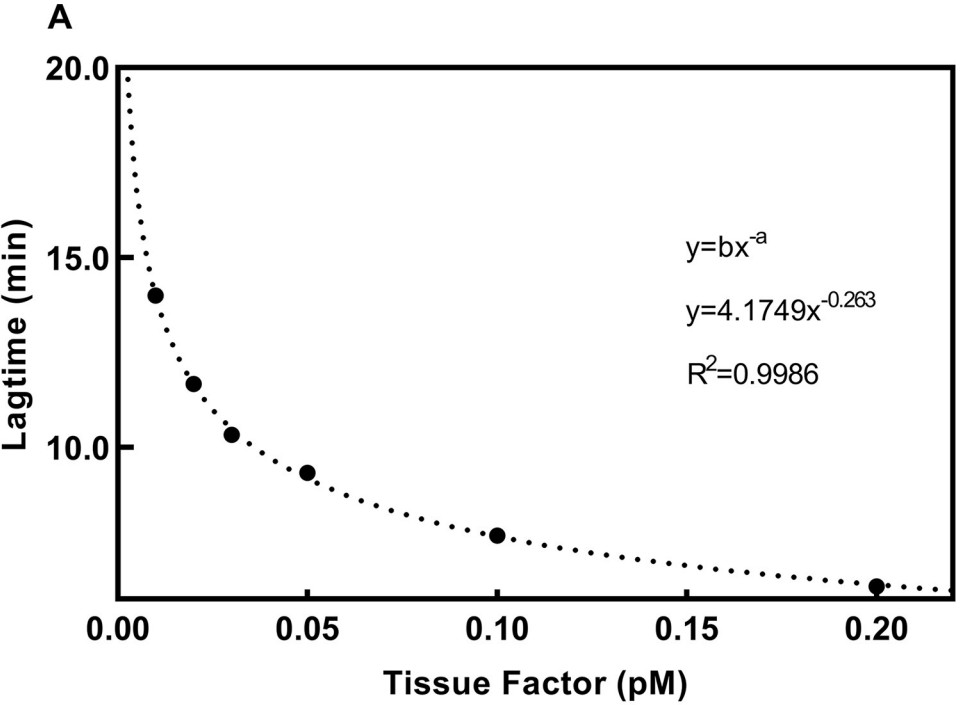

**B**

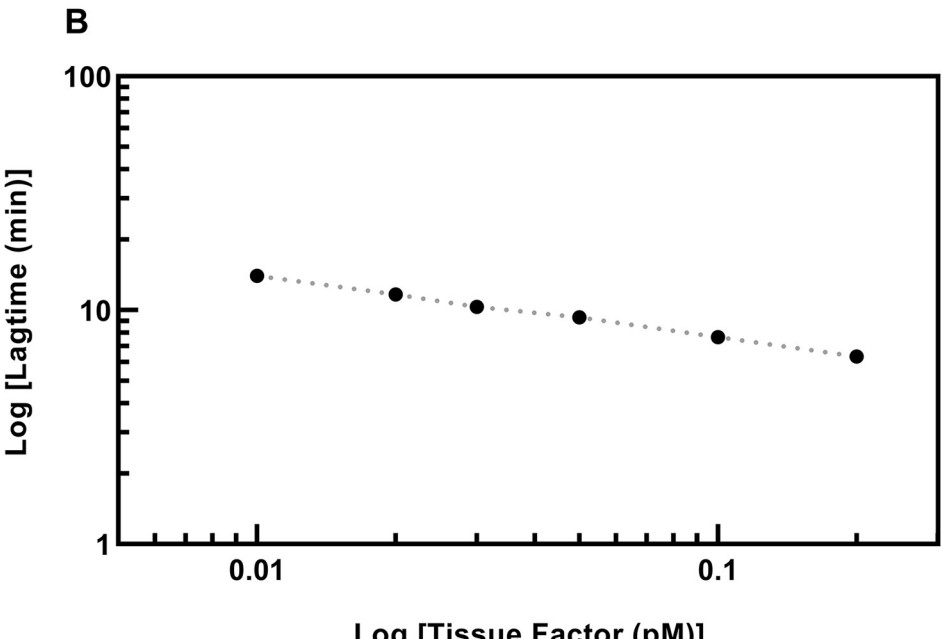

**Fig 2. A calibration curve for TF concentration.** A shows a calibration curve for the TF concentration vs the Lagtime (and the resulting equation), and B shows that the relationship is linear in a log/log system.

Peak, LT was almost unchanged at these higher concentrations of PPL, although slightly shorter at a concentration of 10 fM TF, where the shortened LT corresponds to measurements of 11 fM at 5 μM PPL, 13 fM at 6 μM, and 15 fM at 8 μM PPL. Therefore, when measuring TF using EVs with procoagulant phospholipids, a potential increase of PPL concentrations has a

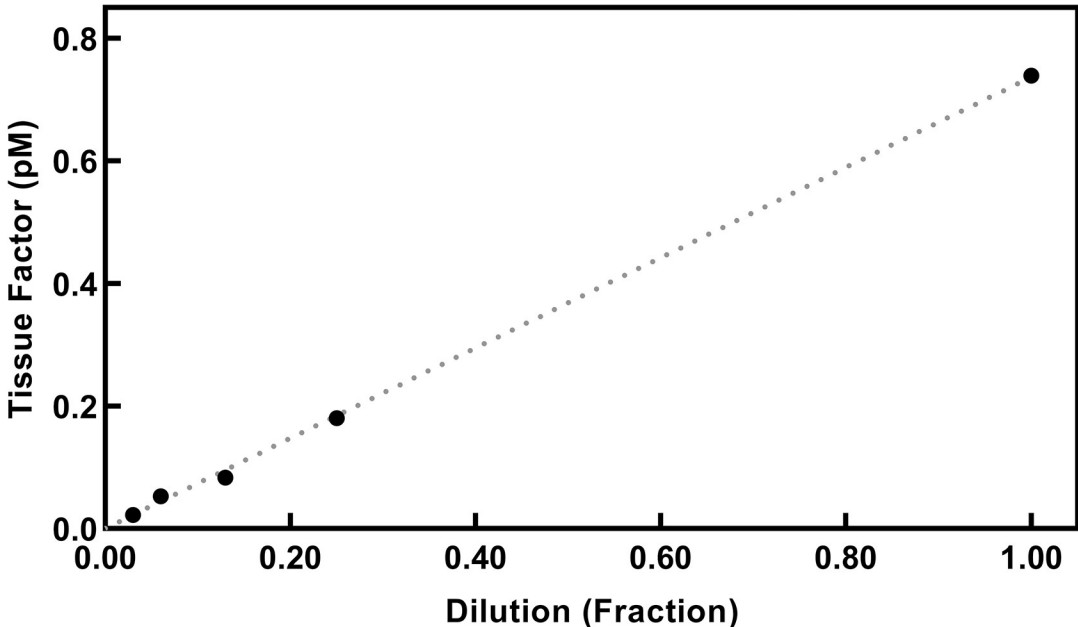

**Fig 3. Linearity of dilutions of TF in an LPS stimulated sample.** The fig depicts the measured TF concentrations of dilutions of a plasma (having a concentration of 0.8 pM TF) from a blood sample which has been stimulated with LPS. The dilution curve is linear.

very small effect on the measurement, but very low concentrations of TF may be overestimated.

## Variation

In the final optimized method, the coefficient of variation (CV) of LT for the duplicates of the calibrators (i.e. $CV_{within}$) were 3,7% for 0.01 pM TF, and 1.9–3.4% for the calibrators between 0.02 pM and 0.5 pM. The total variation for all the calibrators during 3 months analyses were 6–8%. The total variation ($CV_{total}$) of measurements of TF activity at a very low TF concentration was below 20% when measured on three different plates (duplicates on each). At the higher level in LPS stimulated samples CV was below 10% determined in the same way. It has been shown earlier that the major variation originates from the isolation of EVs [10]. We did not test this extensively but from three centrifugations of 4 samples having a concentration of 200–300 fM, the variation was 10–33%. From three centrifugations of 4 samples at a concentration of 1–4 fM, the variation was 50–100%, a considerable variation, but demonstrating that very low concentrations can repeatedly be measured as very low concentrations.

## Discussion

We have optimized a method to measure TF activity based on thrombin generation. It is important to avoid contact activation and to add antibodies against TFPI to get reproducible results of low TF concentrations.

Loeffen [25] has described the difference of TG measured in various different sample tubes, and in accordance with previous findings [26] we also found that Monovette tubes have a much lower contact activation than Vacutainer tubes. However, Monovette tubes still induce some contact activation because the thrombin generation cannot be inhibited totally by aTF (Fig 1A). Addition of antibodies against FXIIa which formerly has been used to avoid contact

**A**

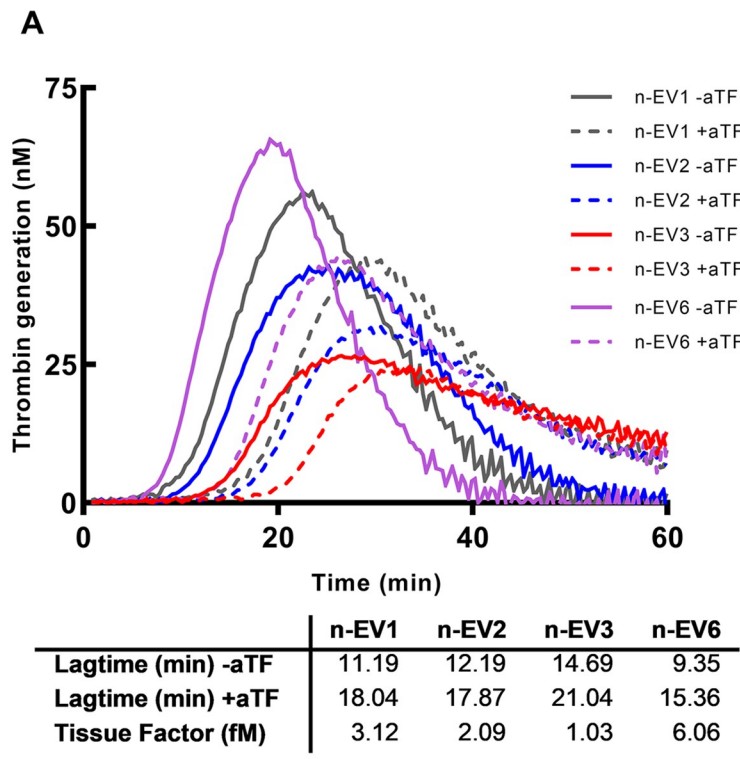

|                        | n-EV1 | n-EV2 | n-EV3 | n-EV6 |
|------------------------|-------|-------|-------|-------|
| Lagtime (min) -aTF     | 11.19 | 12.19 | 14.69 | 9.35  |
| Lagtime (min) +aTF     | 18.04 | 17.87 | 21.04 | 15.36 |
| Tissue Factor (fM)     | 3.12  | 2.09  | 1.03  | 6.06  |

**B**

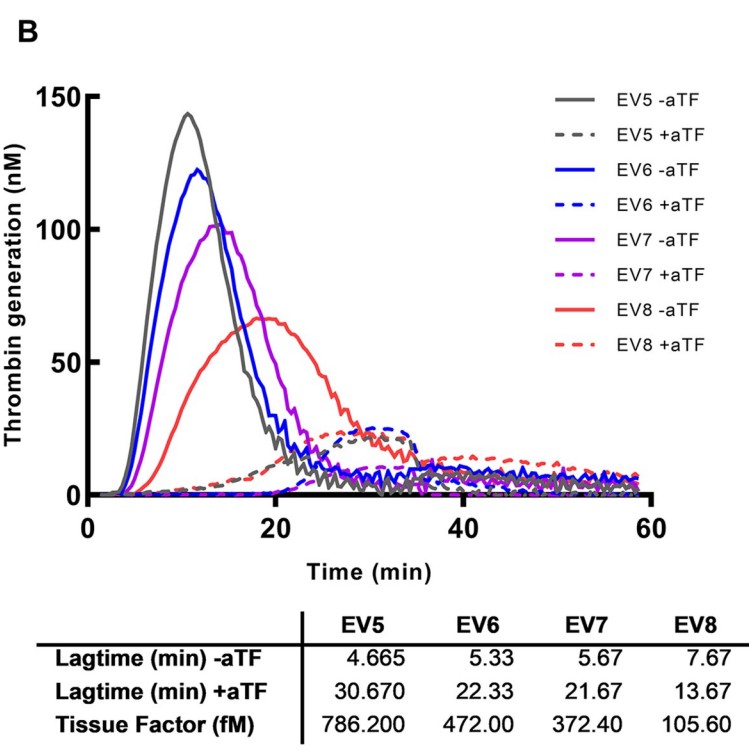

|                        | EV5     | EV6    | EV7    | EV8    |
|------------------------|---------|--------|--------|--------|
| Lagtime (min) -aTF     | 4.665   | 5.33   | 5.67   | 7.67   |
| Lagtime (min) +aTF     | 30.670  | 22.33  | 21.67  | 13.67  |
| Tissue Factor (fM)     | 786.200 | 472.00 | 372.40 | 105.60 |

**Fig 4. Measurements of TF in samples.** A depicts the thrombin generation measured in EV samples from 4 normal controls having 1 (red, n-EV3), 2 (blue, nEV2), 3 (black, n-EV1) and 6 fM TF (purple, n-EV6). The EVs are concentrated from 7500 µl plasma sampled resuspended in 300 µl SP before mixing with SP in the TG analysis. The dashed lines show the thrombin generation in samples with aTF added, i.e. there is a small TF independent activation in these samples. B shows the thrombin generation in four LPS stimulated blood samples. EV-TF concentrations (EV5-EV8) were between 300 and 800 fM. Below the figures are shown LT and the corresponding TF activity.

activation [18, 21] was not sufficient. Therefore, we tested addition of Corn Trypsin inhibitor (CTI) at a concentration of 18.3 µg/mL to the blood sample, Fig 1B. With this addition we saw no activation when no TF was added, and the differences of LT between 0.01 pM and 0.1 pM was much more marked. According to Luddington and Baglin [19] this concentration is sufficient to inhibit contact activation, and a higher concentration only had a small effect (S3 Fig). Therefore, addition of 18.3 µg/L was chosen, also because of the expense of CTI (CTI additions are costly). The basis for the TG activity is a SP. Normal plasma is usually produced from

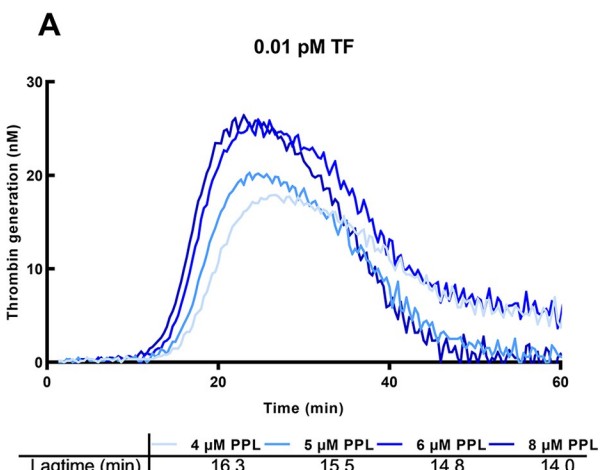

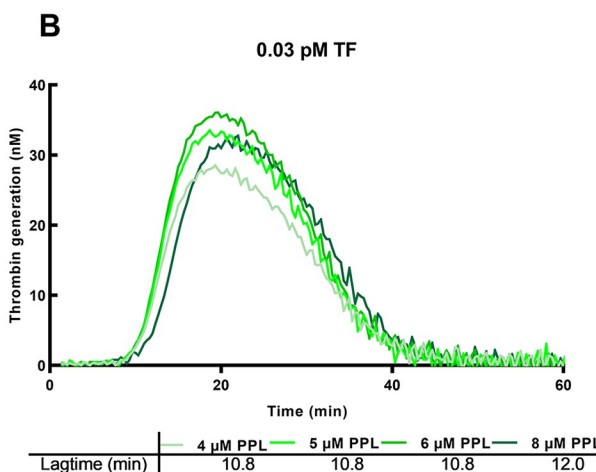

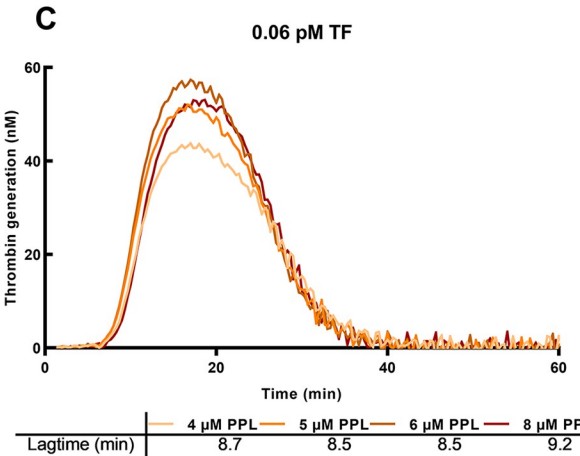

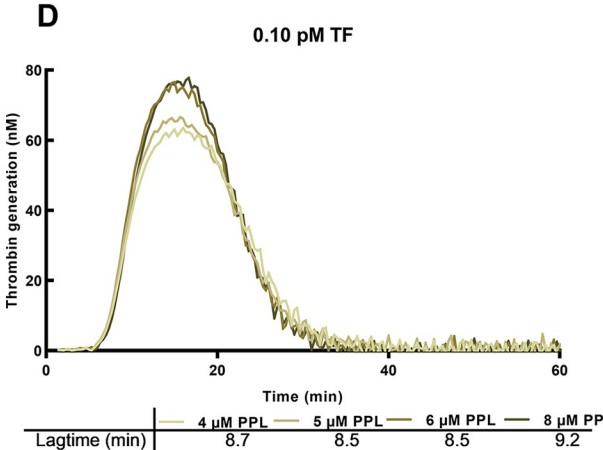

**Fig 5. Effect of an increased concentration of phospholipid.** Thrombin generation in samples where the phospholipid (PPL) concentration is increased from the normal level (4 µM) to 5, 6 or 8 µM. The TF concentrations were from 0.01 to 0.1 pM. Below the figures are shown LT. At the higher concentrations of TF the effect of PL is negligible whereas there is a small effect at the lower concentrations.

samples from at least 20 normal persons. We used a lower number of persons, but we always used the same SP when we made comparisons. Although the differences between different stocks of the SP that we used actually were not marked (the TF activity is calculated from the calibration curves), it is probably preferable to use the same SP to a particular project where TF activity should be measured, but not mandatory.

TG has been used before to estimate the presence of TF. Bidot et al. [14] showed in 2008 that TG, where EVs from various patients were added to a SP, could differentiate between patients with recurrent VTE, patients with a single event of VTE, and controls. LT was shortest and peak highest in patients with recurrent VTE, intermediary in patients with a single VTE and LT was longest and the peak smallest in controls. They concluded that EVs had a procoagulant effect, probably because EVs contained TF, but they did not quantitate TF. Debaugnies [15] compared cancer patients and controls using TG and found a difference but did not quantitate the TF concentration. Ollivier [13] described a method to measure TF based on TG where plasma samples with CTI added could detect a concentration of TF on 0.025 pM and higher although it was not calibrated to estimate the concentrations. They saw an increasing TG when PPL was included in the reaction up to 4 μM. Boknäs et al. demonstrated the importance of CTI especially when the reagent contain PPL [23]. Hellum et al. [16] used TG to detect TF in patients with pancreatic cancer and found that addition of aTFPI improved the signals considerably—we could confirm that this concentration was sufficient to inhibit TFPI (S4 Fig). The present method has included several of these findings in an optimized method. Important points are addition of CTI to avoid contact activation, and addition of aTFPI [16] which increased the signals of the low TF concentrations (shorter LT). The calibration curves are based on the reagents from Stago containing 1 pM TF. aTF were added in a concentration used previously [18] which ensures that the activation is TF dependent, and higher concentrations had no further effect (S2 Fig). This resulted in a linear calibration curve in a log-log system (Fig 2), and dilutions were linear (Fig 3). The procoagulant effect of EVs on TG is because of the presence of TF but may also be promoted by PPL of the EVs. Hemker et al. described that the effect of an increasing concentration of PPL on TG was small at concentrations $> 3$ μmol/L which is the reason for using 4 μmol/L in the reagents for CAT [11]. The effect of higher concentrations in our set-up i.e., final concentrations of 5–8 μmol/L has not been investigated before in the papers measuring EV-TF. This increased concentration only affected LT at very low TF concentrations (Fig 5)–at a TF activity of 10 fM the activity was overestimated 10–50% with PPL concentrations at 5–8 μM, but no overestimation at a TF activity of 30 fM or higher. Therefore, when measuring TF using EVs containing procoagulant phospholipids, a potential increase of PPL concentrations above the 4 μM in the reagent has a very small effect on the measurements, but very low concentrations of TF will probably be overestimated.

We hypothesized that addition of FVIIa might improve the signals because TF could immediately bind to this activated FVII, but it did not improve the method since LT after addition of aTF was also shortened, probably because FVIIa in combination with phospholipids can activate FXa, i.e. a TF-independent activation [5].

Vallier et al. [10] investigated the effect of various centrifugations to isolate EVs and based on this we chose 20,000*g for one hour as we have also used before [27]. At very low EV-TF concentrations a TF-independent TG was seen in the presence of aTF (Fig 4A). Nubouossie et al. [24] described recently that red blood cell EVs could activate the coagulation probably via an activation of prekallikrein and FIX and this activity was not inactivated by CTI but could be inhibited by soybean trypsin inhibitor [24]. However, addition of soybean trypsin inhibitor was not successful because SBTI inhibited this activity as well as the TF activity (S7 Fig). But the apparent TF concentration from this activity induced by EVs (i.e. when aTF was added)

was subtracted from the activity read on the calibration curve for the sample in the absence of aTF, and it only reduced the TF concentration to a lesser extent at the low level and negligibly at the higher levels (Fig 4B). As mentioned, the premise for this subtraction is additivity of the TF-dependent and TF-independent activity of the EVs. This way of adjusting for the TF-independent activity may not be completely correct, and especially at low TF activities this may affect the result, whereas at higher TF concentrations it is less important. In combination with the potential effect of PPL from EVs when measuring low levels of TF, this reduces the accuracy of these measurements. The results indicate a very low level of TF activity and although the measurements are rather reproducible the exact activity has an inherent uncertainty. At higher levels as found in LPS stimulated samples the interference from these potential sources of error have virtually no importance for the result. Thus, when using the method to estimate the TF activity in procoagulant patients with an increased level of TF, the method is reliable.

The variation of the method calculated from the duplicates of calibrators was low: a very low $CV_{within}$, and a quite low $CV_{total}$. The total variation of measuring TF activity of samples based on CV calculated from duplicates was reasonable and comparable to the results of Hellum et al. [16]. The isolation procedure of EVs, however, has an inherent high degree of variation as shown by Vallier et al. [10] who recently published an improved version of a previous method to measure TF concentration based on FXa formation. They could detect TF in normal donors which had levels of $7 \pm 4$ fM, a similar or slightly higher level than our values. However, comparison is difficult when the calibrant differs. The CVs were not very different from the present method, probably slightly lower but it is difficult to compare CVs at different levels. A drawback of this method is the duration of the assay because of rather long incubation periods and the possibility of some TF-independent activity as stated by Østerud et al. [5]. The method recently published by Østerud et al. [5] is rapid but probably less sensitive since they found no activity of TF in normal controls. It had a lower CV than the present one and it is faster. The advantage of our method is that it is a one-stage method using commercially available reagents and the CAT system present in many research laboratories, and, therefore, simpler to establish.

In conclusion, we have established a method to detect TF activity based on thrombin generation with a high sensitivity and a reasonably low variation.

## Supporting information

**S1 Fig. Effect of aFXII and CTI to inhibit the contact activation.** Addition of a FXII increases LT by some minutes, but addition of CTI to the blood increases LT substantially. The bars show mean and SD of 4 plasma samples.
(TIF)

**S2 Fig. Effect of increasing the concentration of aTF.** The figure shows the ratio between LT in the presence and the absence of antibodies, when TG is activated by 0.05 and 0.1 pM TF. I. e., the more the antibodies inhibit TF the higher ratio will be expected. It appears that increasing aTF to higher concentrations do not increase the ratio indicating that 7.8 µg/mL is sufficient to inhibit the TF activity. The bars show mean and SD of 4 plasma samples.
(TIF)

**S3 Fig. Comparison between 18.3 µg/mL and 36.6 µg/mL CTI.** Calibration curves (i.e. LT vs TF concentration) using 0.01–0.10 pM TF to plasma from blood samples where CTI was added in the two concentrations.
(TIF)

**S4 Fig. Effect of addition of aTFPI.** Addition of 100 µg/L or 150 µg/L aTFPI reduce LT in samples with 0.1, 0.2 or 0.3 pM TF considerably but no difference between 100 and 150 µg/mL. The bars show mean and SD of 4 plasma samples.
(TIF)

**S5 Fig. Effect of addition of FVIIa.** Addition of FVIIa reduces LT but it also reduces LT in the same sample in the presence of aTF, i.e. that the effect is not a TF dependent activity.
(TIF)

**S6 Fig. Measurements of TF in samples.** The same figure as Fig 4, but including all seven normal volunteers with TF activity of 1–6 pM. Below the figure is described LT in the presence and absence of aTF and the corresponding TF activity.
(TIF)

**S7 Fig. Effect of soybean trypsin inhibitor.** SBTI in various concentrations was added to a sample with 1 pM TF. It appears that the higher concentrations of SBTI inhibit the activity in samples without aTF as well as in the presence of aTF.
(TIF)

## Author Contributions

**Conceptualization:** Søren Risom Kristensen.

**Data curation:** Søren Risom Kristensen, Jette Nybo.

**Formal analysis:** Søren Risom Kristensen, Jette Nybo.

**Methodology:** Søren Risom Kristensen, Jette Nybo.

**Visualization:** Jette Nybo.

**Writing – original draft:** Søren Risom Kristensen.

**Writing – review & editing:** Søren Risom Kristensen, Jette Nybo.

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
