## [Decision Letter · Decision Letter 0]

24 Feb 2023

PONE-D-23-00356

A sensitive tissue factor activity assay determined by an optimized thrombin generation method

PLOS ONE

Dear Dr. Kristensen,

Thank you for submitting your manuscript to PLOS ONE. After careful consideration, we feel that it has merit but does not fully meet PLOS ONE’s publication criteria as it currently stands. Therefore, we invite you to submit a revised version of the manuscript that addresses the points raised during the review process.

We look forward to receiving your revised manuscript.

Kind regards,

Heather Faith Pidcoke, MD, MSCI, PhD

Academic Editor

PLOS ONE

Journal Requirements:

Additional Editor Comments:

The reviewers have spent a considerable amount of time providing careful review and commentary on this manuscript. They have submitted detailed feedback and recommendations which are aimed at improving the presentation of this work. If the authors address these comments, both reviewers are of the opinion that this paper would constitute a valuable contribution to scientific literature. Please address their comments by either improving the manuscript as suggested or by adding a justification for the original choices. 

Please note that the reviewers disagreed on whether the manuscript is presented in an intelligible fashion and written in standard English. Reviewer 1 provided detailed commentary on improving the organization of the paper and the presentation of the data - please address these comments if you choose to submit a revised version of the manuscript.

This editor and both reviewers are of the opinion that the statistical analysis requires improvement. Please consider requesting input from someone trained in statistical analysis of scientific data.

Reviewer 1 further makes the recommendation that the authors add a comparison to the Mackman lab in-house assay, published by Hisada et al. While this is sound advice and would strengthen this publication, it is not a requirement for resubmission of the manuscript.

Reviewers' comments:

Reviewer's Responses to Questions

**Comments to the Author**

1. Is the manuscript technically sound, and do the data support the conclusions?

Reviewer #1: Yes

Reviewer #2: Partly

2. Has the statistical analysis been performed appropriately and rigorously? 

Reviewer #1: No

Reviewer #2: No

3. Have the authors made all data underlying the findings in their manuscript fully available?

Reviewer #1: No

Reviewer #2: No

4. Is the manuscript presented in an intelligible fashion and written in standard English?

Reviewer #1: No

Reviewer #2: Yes

5. Review Comments to the Author

Reviewer #1: The authors present a publication titled “A sensitive tissue factor activity assay determined by an optimized thrombin generation method”. The purpose of this publication as stated by the authors was to “establish a sensitive method to detect low levels of tissue factor (TF) based on the calibrated automated thrombogram (CAT) method”. This is important and relevant research to have simple measurement of the major initiator of the coagulation cascade. The study strengths are to use commercial grade reagents and a well-known method to rapidly determine tissue factor activity in plasma samples. This test would hopefully be an important step to bring measure of tissue factor from the research bench using chromatography assays to more of a commercial test. Despite these strengths, the paper is poorly written, organized, and fails to provide the data to be sufficient for publication in current form. Please see general comments and specific comments below:

The first main weakness in this paper is poor presentation to support their findings. Although the authors present graphical examples of the seven donors tested they fail to supply table listing the demographics of these donors (all men? how old?). Another table in the paper or supplemental data should list the computed CAT values and the CVs of each of the runs. This prevents a detailed review of the papers finding which is hard to get from the graphs. Without this date it is difficult to determine the CVs independently. Also help in understanding if additional donor testing is needed to reduce the CV to less than 10%, which is needed for high-quality research assay. Therefore, it is recommending the authors present this data and consider increasing the number of donors to reduce the CV value. As there are great number of tests out their increasing the number of donors documenting the sensitivity and specificity of the test is important. This is needed despite previously published low donor tested Hellum et al (only three donors) and Osterund et al (did not report how many). Furthermore, sex and age and other pertinent demographic variables are important for rigor and reproducibility in research. Although, Scully et al (J Thromb Thrombolysis, 2009) documented no difference in monocyte expression of TF between men and women. The NIH states that we should always consider sex, race, and other characteristics as biological variable. . Lastly, the authors should report sensitivity and specificity in predicting a TF expression in a donor using their test, which is necessary to compare tests.

Furthermore, as many studies have been published to measure tissue factor activity, it would be good for the authors to review the publications, cite from Mackman group, and discuss other methods in their paper including:

1. Mackman N, Sachetto ATA, Hisada Y. Measurement of tissue factor-positive extracellular vesicles in plasma: strengths and weaknesses of current methods. Curr Opin Hematol. 2022 Sep 1;29(5):266-274. doi: 10.1097/MOH.0000000000000730. Epub 2022 Jul 11. PMID: 35852819.

2. Hisada Y, Mackman N. Measurement of tissue factor activity in extracellular vesicles from human plasma samples. Res Pract Thromb Haemost. 2018 Nov 20;3(1):44-48. doi: 10.1002/rth2.12165. PMID: 30656275; PMCID: PMC6332748.

The first is excellent review that should be cite and evaluated in reporting a novel sensitive test. The second paper (Hisada et al) is the publication of the Mackman lab in-house assay which would strengthen this publication if the authors were to add a comparison to this test. This reviewer recommends repeating the study to compare these methods. Otherwise, this paper will not have the impact it needs to move toward a commercial test. Lastly, explain why an increased phospholipids would increase the lag time. As Mackman et al states above phospholipids that come from cellular activation (especially phosphatidylserine) activate cryptic TF to become active TF. This would lead to a decrease in lag time. Please explain why you found an increase in lagtime or was it not significant because of high CV.

Overall, figure legends should be at the end or connected to figures not in the text. All acronyms need to be spelled out. Overall, methods are very disorganized and large sections are missing compared to recent Osterund et al paper included NTA analysis and other details on the research process.

ABSTRACT:

Line 22: Tissue factor concentration is different than activity as you are measuring only activity it's important to mention that. You are not measuring the total amount of tissue factor, the active vs. non-active (aka cryptic).

Line 23: Are you determining the sensitivity and specificity of your assay or are you studying the CV?

Line 25: Please correc that the assay is based on “Thrombin generation”, it's based on :”lag time” to thrombin generation

Line 28: define lag time

Line 30: define calibrators

Line 31: This should be in methods: “Contact activation of the coagulation system was avoided using CTI plasma samples in Monovette tubes.”

Line 34: There is no mention on the methods of how you measured EV concentrations. Clarify EV up concentrated as this is not a standard term

Line 35: “However, the step with isolation of EVs have a higher inherent CV.” This is not appropriate for abstract but should be in the limitations.

Line 37: Precise would be all CV less than 10% for research-based assay and less than 5% for diagnostic assays. I would remove this word.

INTRODUCTION:

Line 42: Tissue factor is the PRIMARY coagulation factor that activates the coagulation system, please see papers on cell-based coagulation

Line 43: Do you mean increased expression of tissue factor?

Line 45: What do you mean present inside vascular walls, do you mean endothelial cell?

Line 48: What do you mean the measurement of TF, protein, or activity?

Line 61: Please discuss the Yohei paper as referenced above

Line 62: Please clarify the purpose of the hypothesis is it to determine the sensitivity of the test or accuracy (CV) and is it for protein expression or tissue factor activity

MATERIALS AND METHODS:

Line 67: Please make materials section into one paragraph

Line 81: Most TF activity assays discard the first three ml of blood draw, why was this not followed?

Line 95: Why did you do CAT measurements in duplicates? The thrombinoscope software recommends triplicates.

Line 99: Spell out all acronyms…ETP

Line 104: Please state the standard dilution of the MP reagent, not spelled out acronym

Line 107: Please clarify that the incubation with aTF was done before CAT or is it wrong section of the methods

Line 113: Why no sensitivity or specificity calculation as you did a test on TF EV from donor blood.

RESULT

Line 119: This is methods part and need to be moved up and explained.

Line 145: Watch out for acronyms like fig. please spell out

Line 127-129: This explanation needs to go to the methods.

Line 131 to 135 Please put into figure legend (put all figure descriptions into a legend)

Line 137 to 139: This is discussion and needs to be betterl explained

Line 149 to 151: Goes into methods

Line 151: How were EVs concentration measured, this needs to be put in methods

Line 153: Were not all samples treated the same? Why are higher ones achieved with LPS?

Line 153: we need a table describing the age, gender, and others to help repeat study

Line 165: Phospholipids usually activate TF, please explain in discussion why would lag time be prolonged with increased phospholipids

Line 175: a table needed to list all of this data, the CV and other can be in supplemental.

Line 179-181: As the variation is high it would be good to test vs. a published assay (Hisada et al) for reference.

For graphs and figures be sure your legends are present and spelled out in figures.

DISCUSSION:

Line 195: Can you comment on the CTI differences between 18.3 and 36.6 ug/mL as the data is not presented. Can it be a difference because the first 3 ML of blood was not discarded as performed in other studies.

Line 198-201: Very long sentence, please revise in two or three sentences.

Line 201-208: This is a list of papers with no relation of their findings with the results

Line 208-211: Rephrase to say we chose the previous method to include in our study as..

Line 221: Evs are made up of phospholipids. How do you distinguish that one of the donors EVS do not contribute the phospholipids to override the system. How many of the EVs were leukocytes vs. platelets? Platelet EVs do not contain TF but are more numerous that leukocyte

Line 224-225: Should this be iits own paragraph?

Line 230: Did you do test with soybean trypsin inhibitor? Nothing was reported in results. please include the data in supplemental.

Line 236-237: Please see line 224-225

Line 238: Please report your CVs for all values and then mention the ones for Vallier so we can know the actual difference. If your CVs are not less than 10% for lower TF, then how can you call this improved?

Line 243: Is it possible they found no TF in normal controls because they discarded the first 3 mL?

Line 246: This conclusion is too short and does not have a limitation section.

Reviewer #2: Summary: The authors present an interesting study detailing a novel method for determination of tissue factor (TF) concentration utilizing a commercially available thrombin generation assay (CAT; STAGO). Development of such an assay would make quantification of TF more readily available for both clinical and research investigations.

While the overarching objective of the study, to measure TF based on TG, is clear, the specific sub-aims/experimental procedures implemented are not clearly defined in the introduction or methods sections. There is no study hypothesis. Currently, the manuscript is missing a clear delineation of Methods/Results/Discussion as portions of each of these sections are presented in different areas/not clearly in a sequential order. The reader is left to go back and forth between sections for clarity at several points. It would be useful to have 1-2 sentences in the introduction clarifying how the authors planned to go about developing and evaluating the TG based TF detection method, and what conditions/variables will be assessed in this investigation – as this is not readily apparent. Further, the statistical methods are lacking. The sample sizes are not clearly reported, and only representative data displayed. It is not stated whether a power analysis was performed or whether the study was accurately powered to assess each of the outcome measures investigated. At various points throughout the manuscript the authors mentioned that they evaluated an experimental condition – but there is no data shown to support this. The manuscript could strongly benefit from a detailed Supplemental Methods and Supplemental Results document so that all data and experimental procedures referenced are accounted for and presented. It would be important to perform a comparative study to assess TF using the TG method developed in this study alongside other reference methods, prior to making claims that this is an “established” method.

Additional comments:

Methods:

How many healthy donors were included in the donor pool for preparation of the SP? Likely the experiment should be repeated additional times with SP prepared from separate donor pools. Likewise, may be important to repeat the study using SP prepared from patients with pathologies subject to TF investigation.

The method section could benefit from a table that indicates what additives /components were utilized in the sample mixtures for each of the experiments performed (ex. aTFPI, SP, aTF, EV suspension, CTI, etc.)

How was the concentration of CTI to be used decided upon? *Update - this is mentioned later in discussion but should be clarified in the methods earlier and cited appropriately. Were other concentrations tested by these authors? And where are those results reported?

Throughout methods and results it is difficult to ascertain how many times the experiments were repeated. The statistical analysis section in the methods states that representative data is presented, but throughout the manuscript the sample sizes are never clarified - it is only mentioned that "several" replicates were performed. The sample sizes should be clearly stated. Additionally, was a power analysis performed prior to the study? Or how were the sample sizes decided upon? Was the study accurately powered at each phase to draw the conclusions reported?

Results

Are there reference data or experimental data that could be added to a supplement to show the variability in LT when aTFPI was applied versus when it was not? How was the concentration of aTFPI decided on?

Results line 137-138 states, “Ten pM [FVIIa] had a small effect which increased at higher concentrations, but an effect was also seen in samples with aTF added….” – Where are these results? Could they be provided in a supplement? What did the authors define as “a small effect”/what was the outcome of interest here? LT? How many times was this experiment repeated?

In Fig 2 and 3, again it is not clearly stated how many replicates were performed to generate the TF concentration vs LT calibration curve; or how many replicates were performed when assessing linearity of the dilutions of TF in the LPS stimulated samples.

In Figure 4, thrombin generation in EV samples from 3 normal controls having 1-3 fM TF is shown – but earlier in the results it states that levels of TF in the normal controls ranged from 1-6 fM. Why are results for 4-6 fM not shown? Why do the authors speculate that LT was prolonged with escalating TF concentration in Fig 4A? What were the actual TF concentrations for EV5-8 in Fig 4B?

In Figure 5, the abbreviation “PPL” is used in the figure caption and manuscript, but the abbreviation “PL” is used on the figures themselves – this should be standardized.

Throughout the results, it may be useful to provide the numerical LTs – it is difficult to ascertain the LTs and variability in LT from the figures and CV alone, especially when no sample size is provided. The ETP/Peak are more clearly depicted in figure format; however, if LT is the sole outcome of interest it may be beneficial to report mean results in a table format with standard deviations and sample sizes. This information can be obtained in part from Figure 2A – however there are no error bars or samples sizes included in the figure to indicate how many replicates were performed to achieve a lagtime result at each TF concentration.

Discussion

The discussion mentions that additional CTI concentrations were evaluated – where are these results/could the be provided in supplement?

Lines 211-218 provide important information that should be explained in the methods section, rather than in discussion.

Line 224-225 Where are these results shown?

Line 230-233 Is this referring to the present study? If so, were these results reported?

Further discussion on how the presented novel method is potentially superior to currently available methods would be important. What are the next steps/future directions for this work?

6. PLOS authors have the option to publish the peer review history of their article (what does this mean?). If published, this will include your full peer review and any attached files.

Reviewer #1: **Yes: **Andrew D. Meyer, MD, MS

Reviewer #2: No

---

## [Author Response · Author response to Decision Letter 0]

6 Apr 2023

We have uploaded a "Response to reviewers" with an answer to all the points raised

---

## [Decision Letter · Decision Letter 1]

19 Jun 2023

PONE-D-23-00356R1A sensitive tissue factor activity assay determined by an optimized thrombin generation methodPLOS ONE

Dear Dr. Kristensen,

Thank you for submitting your manuscript to PLOS ONE. After careful consideration, we feel that it has merit but does not fully meet PLOS ONE’s publication criteria as it currently stands. Therefore, we invite you to submit a revised version of the manuscript that addresses the points raised during the review process. Please note that in response to your concerns about the reviewer comments regarding your original submission, we have requested the evaluation of a third reviewer whose paper you have cited in your submission. This reviewer (Reviewer #3) has provided comments regarding your work. Please address these comments in full prior to re-submitting your manuscript.

We look forward to receiving your revised manuscript.

Kind regards,

Heather Faith Pidcoke, MD, MSCI, PhD

Academic Editor

PLOS ONE

Journal Requirements:

Reviewers' comments:

Reviewer's Responses to Questions

**Comments to the Author**

1. If the authors have adequately addressed your comments raised in a previous round of review and you feel that this manuscript is now acceptable for publication, you may indicate that here to bypass the “Comments to the Author” section, enter your conflict of interest statement in the “Confidential to Editor” section, and submit your "Accept" recommendation.

Reviewer #1: All comments have been addressed

Reviewer #3: (No Response)

2. Is the manuscript technically sound, and do the data support the conclusions?

Reviewer #1: Yes

Reviewer #3: Partly

3. Has the statistical analysis been performed appropriately and rigorously? 

Reviewer #1: Yes

Reviewer #3: I Don't Know

4. Have the authors made all data underlying the findings in their manuscript fully available?

Reviewer #1: Yes

Reviewer #3: Yes

5. Is the manuscript presented in an intelligible fashion and written in standard English?

Reviewer #1: Yes

Reviewer #3: Yes

6. Review Comments to the Author

Reviewer #1: (No Response)

Reviewer #3: It would have been great to have a reliable sensitive and specific assay for the measurement of TF activity in blood/plasma. However, the present assay is not any easier and cheaper assay for TF activity than the present assays available today. It is rather complicated as the EVs must be isolated, and then they are added back to a pool of plasma from different individuals. To avoid any plasma contribution to thrombin generation, they are adding CTI to block the intrinsic pathway and aTFPI to avoid plasma inhibition of TF activity, both expensive reagents.

In the abstract, line 37, the authors state: “addition of PPL only increased lagtime slightly at very low concentrations of TF showing--” How can this be as they in the section 231-237, and Fig 5, shows that TG is enhanced with adding larger amounts of PPL, and LT is slightly shorter at a concentration of 10 fM TF. This does not make sense and should be corrected.

It looks like detection of TF activity in plasma of healthy individuals, is the major goal of this study. To detect the TF activity, they concentrate the EVs up to 25 times, compared to the level of EVs in the plasma. This gives rise to very high levels of PPL that will cause more non-specific TF activity. But what else can be associated with the large amounts of EVs? There are several other procoagulant activities associated with EVs as for example FVIIa, FVa, FIXa and FXIa, that may contribute to shorter LT, especially at low concentrations of TF. I do not believe that subtracting values obtained with aTF from results without aTF, is giving the right answer for the presence of TF activity or not. As the authors themselves states in lines 235-237: “Therefore, when measuring TF using EVs with procoagulant phospholipids, a potential increase of PPL concentrations has a very small effect of the measurement, but very low concentrations of TF may be overestimated”. I agree and believe that the detected TF activity in this assay system in EVs of healthy individuals, has nothing to do with TF activity. To disprove this, they must rule out the contribution of other procoagulant activities associated with highly concentrated EVs in the generation of thrombin.

FVIIa has different ways to mediate thrombin generation in plasma, and to assure that FVIIa is not involved in the generation of thrombin in the present assay in the absence of TF, they should use FVIIai to block the extrinsic pathway.

The authors are focusing on the detection of very low concentrations of TF in plasma to be used for analyzing samples from venous thrombosis patients. In my opinion, procoagulant EVs from activated cells expressing large amounts of phosphatidylserine, play a more important role in thrombin generation in venous thrombosis than the traces of TF detected. Only EVs from plasma of certain cancer patients may have TF levels that can be associated with their events of thrombosis.

Some minor points with strange sentences:

Line 127: “In seven normal volunteers TF activity was measured in plasma and in plasma from LPS-stimulated blood”. How do you define a normal volunteer? TF activity was not measured in plasma but in EVs isolated from plasma.

Line 208: The levels in 7 normal persons.

Line 214: “marked samples with a low TF activity which were concentrated quite much (i.e. a high amount of EVs in”

Line 221: “without aTF, so this non-TF activity did not interfere much with the result”

7. PLOS authors have the option to publish the peer review history of their article (what does this mean?). If published, this will include your full peer review and any attached files.

Reviewer #1: **Yes: **Andrew D Meyer, MD, MS

Reviewer #3: No

---

## [Author Response · Author response to Decision Letter 1]

6 Jul 2023

6. Review Comments to the Author

Reviewer #1: (No Response)

Reviewer #3: It would have been great to have a reliable sensitive and specific assay for the measurement of TF activity in blood/plasma. However, the present assay is not any easier and cheaper assay for TF activity than the present assays available today. It is rather complicated as the EVs must be isolated, and then they are added back to a pool of plasma from different individuals. To avoid any plasma contribution to thrombin generation, they are adding CTI to block the intrinsic pathway and aTFPI to avoid plasma inhibition of TF activity, both expensive reagents.

R: It is correct that it is not very simple or cheap – and this has certainly not been stated in the paper. EVs must be isolated as in other assays for TF activity, but the advantage is that you can use thrombin generation (or CAT) which many researchers have admittance to do. Unlike the two most used methods (as described in the Introduction) you do not have to buy several different reagents which is not always easy and uncomplicated (which I have learned from a long life in the lab). In this respect it is more straightforward to use the CAT technique with few modifications, and in this way it is an alternative to other methods. An optimized method for TF activity measurement based on CAT has not been described before.

In the abstract, line 37, the authors state: “addition of PPL only increased lagtime slightly at very low concentrations of TF showing--” How can this be as they in the section 231-237, and Fig 5, shows that TG is enhanced with adding larger amounts of PPL, and LT is slightly shorter at a concentration of 10 fM TF. This does not make sense and should be corrected.

R: We have now explicitly described the effect on the estimation of TF activity concentration

It looks like detection of TF activity in plasma of healthy individuals, is the major goal of this study. To detect the TF activity, they concentrate the EVs up to 25 times, compared to the level of EVs in the plasma. This gives rise to very high levels of PPL that will cause more non-specific TF activity.

R: No, it is not very important to measure TF activity in normal, healthy individuals. As stated in the introduction it may be interesting to quantitate TF activity in cancer patients and other procoagulant patients. We have done this in some cancer patients previously and using the Mackman method it is not possible to detect an activity in all the patients. It could, therefore, be advantageously to measure lower concentrations. In their optimized method (Ref. #10 in the paper) they could actually measure TF activity in some of the healthy controls, and we therefore found it interesting that it was also possible with this method if we concentrated the EVs. But we believe that the main application of a method like this will be for patients who have a higher TF concentration. EVs were not concentrated 25 times because in the assay 20 µL of this solution is added to 60 µL SP, as described in Methods – we have now added this in the text. Thus it is concentrated about 6 times compared to normal plasma, and in the reaction it is further diluted to two thirds.

As we have described in the paper, high levels of PPL and FVIIa can activate FX, but this is the reason why we have tested the effect of increasing PPL in the assay which have a minor effect. What is “very high levels of PPL”? I do not think that we have very clear estimations of PPL concentrations of EVs. According to measurements with Zymuphene MP activity assay it should be at or below 10 nM PS equivalents. So even if the EVs are concentrated and only part of PPL in the reagent has PS activity, the amount of PPL from EVs cannot be more than the 4µM that we added.

But what else can be associated with the large amounts of EVs? There are several other procoagulant activities associated with EVs as for example FVIIa, FVa, FIXa and FXIa, that may contribute to shorter LT, especially at low concentrations of TF.

R: As stated, there is not a large amount of EVs in the assay, especially not when we measure higher concentrations of TF. There is no reference to these statements? We are not aware of papers that have estimated the concentration of these factors. It must be remembered that EVs are added to a standard plasma with normal concentrations of coagulation factors so this addition will probably be negligible. Even if it is correct, it will only be a problem at very low TF activities. Otherwise, it will also be a problem in other methods where EVs are added. Perhaps the reviewer means that the EVs have a procoagulant surface? That is exactly the reason why we tested the effect of adding phospholipids (fig 5). And as shown by this experiment, this is only a minor problem at low TF concentrations.

 I do not believe that subtracting values obtained with aTF from results without aTF, is giving the right answer for the presence of TF activity or not.

R: This may be true, but it is a way to try to correct for it. We have now explicitly described the premise for this procedure, and also a possible uncertainty of the estimations because of this. In all the methods for TF activity there is some “blank value” which must be corrected for, and this way of correction may not be true in any of the methods. In any case, it only seems to be a potential interference at a very low TF activity as discussed explicitly in the paper now.

 As the authors themselves states in lines 235-237: “Therefore, when measuring TF using EVs with procoagulant phospholipids, a potential increase of PPL concentrations has a very small effect of the measurement, but very low concentrations of TF may be overestimated”. I agree and believe that the detected TF activity in this assay system in EVs of healthy individuals, has nothing to do with TF activity. To disprove this, they must rule out the contribution of other procoagulant activities associated with highly concentrated EVs in the generation of thrombin.

R: Firstly, it is only when measuring TF activity in normal healthy persons that EVs are concentrated, and it is more relevant to measure higher concentrations in procoagulant persons. Secondly, we have participated in a trial of measurements of TF activity and the present method performed quite well. Making a measurement without and with aTF is usually what has been done to measure TF activity. Furthermore, I can tell that we have used the Mackman-method previously in samples from cancer patients – here we determine an activity with and without aTF. Frequently we got a higher activity with aTF included, i.e. we actually found a “negative concentration”. The present method seems to be much more reliable for lower concentrations. 

I think it is a revealing statement that the measured TF activity has nothing to do with TF activity. We measure an activity which is inhibited by aTF. It may be slightly inaccurate but certainly not completely misleading – this is nonsense. How can aTF inhibit an activation which is not performed by TF?

FVIIa has different ways to mediate thrombin generation in plasma, and to assure that FVIIa is not involved in the generation of thrombin in the present assay in the absence of TF, they should use FVIIai to block the extrinsic pathway.

R: As shown in fig 1B there is no thrombin generation in the absence of TF. We have actually tried to add FVIIa and concentrations up to 10 pM did not result in a shorter lagtime, and 100 pM only slightly. Why should there be a high concentration of FVIIa in the assay? Should it come from the EVs? I cannot find any references describing this. FVIIai will bind to TF and therefore inhibit this activity, but will it block all other reactions for the alleged FVIIa? I cannot find documentation for this. When we get a large difference between samples with and without aTF, it is most likely due to a TF activity

The authors are focusing on the detection of very low concentrations of TF in plasma to be used for analyzing samples from venous thrombosis patients. In my opinion, procoagulant EVs from activated cells expressing large amounts of phosphatidylserine, play a more important role in thrombin generation in venous thrombosis than the traces of TF detected. Only EVs from plasma of certain cancer patients may have TF levels that can be associated with their events of thrombosis.

R: It is not correct that we focus on very low concentrations of TF. We agree that it is mainly interesting in certain cancer patients which we intend to investigate (which is also stated in the Introduction). That is the reason why we have established this method which we find more easy to use than the Mackman method.

We have now expressed that more clearly in the paper

Some minor points with strange sentences:

Line 127: “In seven normal volunteers TF activity was measured in plasma and in plasma from LPS-stimulated blood”. How do you define a normal volunteer? TF activity was not measured in plasma but in EVs isolated from plasma.

R: We have changed it to healthy volunteers. We have changed plasma to EVs. 

Line 208: The levels in 7 normal persons. 

R: We have changed it to healthy controls

Line 214: “marked samples with a low TF activity which were concentrated quite much (i.e. a high amount of EVs in”

R: It has been rephrased

Line 221: “without aTF, so this non-TF activity did not interfere much with the result” 

R: We have changed non-TF activity to TF-independent activity

---

## [Editor Report · Decision Letter 2]

7 Jul 2023

A sensitive tissue factor activity assay determined by an optimized thrombin generation method

PONE-D-23-00356R2

Dear Dr. Kristensen,

We’re pleased to inform you that your manuscript has been judged scientifically suitable for publication and will be formally accepted for publication once it meets all outstanding technical requirements.

Kind regards,

Heather Faith Pidcoke, MD, MSCI, PhD

Academic Editor

PLOS ONE
---

## [Editor Report · Acceptance letter]

11 Jul 2023

PONE-D-23-00356R2 

A sensitive tissue factor activity assay determined by an optimized thrombin generation method 

Dear Dr. Kristensen:

I'm pleased to inform you that your manuscript has been deemed suitable for publication in PLOS ONE. Congratulations! Your manuscript is now with our production department. 

Kind regards, 

on behalf of

Dr. Heather Faith Pidcoke 

Academic Editor

PLOS ONE